# Barriers Faced by Australian and New Zealand Women When Sharing Experiences of Family Violence with Primary Healthcare Providers: A Scoping Review

**DOI:** 10.3390/healthcare11182486

**Published:** 2023-09-07

**Authors:** Jayamini Chathurika Rathnayake, Nadirah Mat Pozian, Julie-Anne Carroll, Julie King

**Affiliations:** School of Public Health and Social Work, Faculty of Health, Queensland University of Technology, Kelvin Grove Campus, Victoria Park Road, Kelvin Grove, QLD 5069, Australia; sra.rathnayake@connect.qut.edu.au (J.C.R.); nadirah.matpozian@hdr.qut.edu.au (N.M.P.); j.macknight-king@qut.edu.au (J.K.)

**Keywords:** domestic violence, primary healthcare, general practitioners, female victims, nurses, midwives

## Abstract

Despite the Australian Government’s attempts to reduce domestic violence (DV) incidences, impediments within the social and health systems and current interventions designed to identify DV victims may be contributing to female victims’ reluctance to disclose DV experiences to their primary healthcare providers. This scoping review aimed to provide the state of evidence regarding reluctance to disclose DV incidents, symptoms and comorbidities that patients present to healthcare providers, current detection systems and interventions in clinical settings, and recommendations to generate more effective responses to DV. Findings revealed that female victims are reluctant to disclose DV because they do not trust or believe that general practitioners can help them to solve their issues, and they do not acknowledge that they are in an abusive relationship, and are unaware that they are in one, or have been victims of DV. The most common symptoms and comorbidities victims present with are sleep difficulties, substance use and anxiety. Not all GPs are equipped with knowledge about comorbidities signalling cases of DV. These DV screening programs are the most prominent intervention types within Australian primary health services and are currently not sufficiently nuanced nor sensitive to screen with accuracy. Finally, this scoping review provides formative evidence that in order for more accurate and reliable data regarding disclosure in healthcare settings to be collected, gender power imbalances in the health workforce should be redressed, and advocacy of gender equality and the change of social structures in both Australia and New Zealand remain the focus for reducing DV in these countries.

## 1. Introduction

Domestic violence (DV) is characterised as a series of behaviours used by a perpetrator to obtain or maintain power and authority over an intimate partner in any relationship, as well as over children and/or siblings with whom they share a similar household or a domestic relationship [1,2]. The most prominent forms of gender-based violence are intimate partner violence, rape, sexual assault and stalking [3,4]. DV is regarded as a violation of women’s rights and has emerged as a major and urgent public health issue [5,6,7,8]. Eradicating violence against women was included in the United Nations’ Millennium Development Goals (in 2000) as well as in the Sustainable Development Goal 5 (Gender Equality) (in 2015) [6,9].

Extant findings demonstrate that DV adversely affects women’s health, overall functioning and well-being—in the short and/or the long term (e.g., quality of life) [5,6,7,8,9,10,11,12]. According to the US Department of Health and Human Services (USDHHS) [13], short-term impacts of DV include injuries, bleeding, miscarriages, unplanned pregnancies, sexually transmitted infections and insomnia. The USDHHS [13] further states that the long-term effects of DV include arthritis, asthma, sleeping problems, migraines, headaches, stress, depression and chronic pain. Furthermore, the immediate and ongoing impacts of DV on women’s health have been identified in a variety of areas, including mental health issues and physical damage, such as bruises, cuts, teeth and gum damage, skin lesions, stillbirths and head injuries. Studies reveal the signs of DV perpetration include harmful behaviours against children and pets, as well as the use of unsafe driving to instil fear and coercion [4,14,15].

Among these complications, the most concerns expressed by Australian women were mental well-being issues [3,11]. DV is significantly associated with mental health disorders) and is a leading cause of death, disability or illness [3,16]. Additionally, DV impacts individuals’ financial status and contributes to poverty, especially homelessness. According to Dillon et al. [17], there is an increasing correlation between DV and homelessness, particularly among women and children. This evidence corroborates with Mission Australia [18], which stated that in 2018 and 2019, 80,000 women sought professional homelessness support services.

### Prevalence of DV in Australia

Although DV is regarded as a critical national health and welfare issue [19] and the most unspeakable crime in Australia [7], there has been an unprecedented rise in violence and harassment against women over the last three decades [3,20]. According to the Australian Bureau of Statistics Personal Safety Survey 2016, an estimated one in six women (over the age of 15) experienced sexual or physical violence from a current or former cohabiting partner, with women being were more likely to encounter violence from a known individual and in their home [20,21]. Nevertheless, the magnitude of DV incidence remains unknown [22].

Between 2014 and 2015, a woman was killed every nine days by her intimate partner in Australia [19]. In 2017, more than 11,000 women between the age of 15 and 34 experienced DV or sexual harassment [19]. Women are more likely to become victims during their reproductive years [23,24,25]. According to Gartland et al. [24], 20–30% of women suffered physical or mental abuse 1–4 years postpartum. A meta-synthesis study reveals that women aged 45 and above are also at risk of family violence, which may lead to the risk of homelessness in old age [7,26]. There is also a higher risk of family and DV during major crises, such as epidemics and natural disasters [27,28]. Moreover, increases in the number of DV incidents and the frequency of victims visiting primary healthcare services intensify the burden on medical practitioners and frontline healthcare providers [29].

The Australian Government and healthcare sector, both at federal and state levels, are striving to take immediate and decisive action on behalf of victims [30,31,32]. As a widespread service provider, the healthcare sector can provide high-quality healthcare and ensure supportive environments are in place both to enable victims to disclose DV lived experiences and to help victims and survivors overcome their issues [9,33]. Despite these efforts, numerous impediments remain within the current settings (both health and social systems) and interventions [10,34]. These impediments may lead female victims to be reluctant in disclosing their lived experiences of DV to primary healthcare workers or general practitioners (GPs) [10].

While the devastating impact of DV on women and those that they care for is well documented, and the extent of the problem across both Australia and New Zealand carefully tracked, the phenomenon cannot be either accurately measured nor treated if women remain reluctant to disclose the problem to frontline healthcare providers. Further, while community workers in the DV space tend to be the ‘safe spaces’ female DV sufferers go to for assistance, there is a call for greater trust building amongst these same women and GPs in particular. Further, there is an established need for clinicians to be better trained at detecting reluctance to share DV experiences with them in private appointments. This scoping review aimed to collate the relevant literature in a bid to generate a cohesive, evidence-based narrative around barriers for reporting DV within clinical settings in a bid to provide this information to those who need it most.

We aimed to provide an updated and focused review of the barriers female victims face in revealing DV experiences to primary healthcare professionals in the clinical setting and private appointments with GPs. This review generated a summary of (i) the reasons why DV victims do not disclose to GPs and primary healthcare professionals, (ii) symptoms and comorbidities that patients present to healthcare providers, (iii) current detection approaches and quality of interventions in the clinical setting, and (iv) finally provides recommendations to generate more effective responses to DV to clinicians specifically.

## 2. Materials and Methods

### 2.1. Scoping Review Research Questions and Objectives

This study aims to answer the following research questions: (i) What are the reasons DV victims do not disclose to GPs and primary healthcare professionals? (ii) What are the comorbidities and symptoms that DV patients present with? and (iii) what are the current methods of detection and interventions in clinical settings. The objective was to combine the findings to provide recommendations to both researchers and clinicians regarding more effective responses to DV.

### 2.2. Data Sources and Search Strategy

A scoping methodology was used to conduct the review and identify the results. Several search strategies were developed during the process to identify the relevant studies. Four primary databases were used, including CINHAL (nursing and allied health database), PsycINFO, Embase and PubMed. The term ‘domestic violence’ was mainly used to identify articles using the synonyms of ‘family violence’, ‘intimate partner violence’, ‘battered women’ and ‘domestic violence victims’. The phrase ‘domestic violence’ and its synonyms (with a truncation mark) were used along with phrases such as ‘barriers to express’, ‘barriers to reveal’, ‘enablers to reveal’ and ‘motivations to reveal’ to identify the relevant articles. Boolean operators were used to expand the results.

### 2.3. Eligibility Criteria

This scoping review included all study designs, including qualitative, quantitative and mixed-method studies. It focused on Australian and New Zealand studies, given that New Zealand has a similar public health service to Australia. Only full-text articles in English were considered and included in the review.

### 2.4. Exclusion Criteria

All editorials, letters to the editor, newspaper articles, thesis reviews, dissertations and articles from low- and middle-income countries were excluded from the scoping review. Additionally, studies that discussed substance use and DV and postpartum depression and DV were not considered. Figure 1 displays the process used, including the inclusion and exclusion criteria.

## 3. Results

### 3.1. Why DV Victims Do Not Disclose to GPs and Other Primary Health Professionals

GPs are the primary healthcare workers who identify DV most frequently during private appointments through assessments and diagnostic processes [35]. There is still much debate and discussion about who discloses (both voluntarily and unwillingly) DV experiences to GPs and reports DV side effects (e.g., addictions, insomnia and wounds in various stages of health) but not the abuse itself [35,36,37]. Studies by O’ Doherty et al. [34], Meuleners et al. [22] and Hegarty et al. [10] report that most DV victims do not trust their GPs as a professional to whom they can disclose their DV experiences and related illnesses and injuries. Further, victims do not accept their GPs as a solution to solve DV-related issues [22,34,38]. Generally, DV victims have reported that they view GPs solely as clinical health practitioners, rather than as counsellors or professional supporters to whom they would reveal such violence [34]. Hence, most victims seek GPs only to treat their injury, wounds or physical harm; they do not want to obtain psychological or social support [22].

Victims also do not disclose these injuries as DV cases or as part of the abuse to their GPs. DV victims are more likely to disclose injuries or physical harm as accidents or falls rather than abuse [39]. The critical case is that abused women do not like to acknowledge that they are in an abusive relationship and are or had been victims of DV [34,39,40]. Some women were unaware that they had become a victim of a perpetrator or that the violence was part of the DV phenomenon [10,39]. Consequently, despite being able to recognise DV symptoms, it is a complex and difficult task for primary healthcare providers to provide support to victims who do not recognise and acknowledge that they are in unhealthy relationships and are at risk of ongoing and worsening abuse [39]. Overall, there is a significantly low rate of DV disclosure to GPs during clinical appointments; even when DV is identified, it remains challenging to discuss with the victims and even more difficult to intervene with sustained success [20,36,37,38,40].

### 3.2. What Symptoms and Comorbidities Do Patients Present to Healthcare Providers?

Evidence shows the prevalence of DV is common among women who visit GPs [36,37,40,41]. However, women tend not to present their DV experiences or symptoms as symptoms of abuse, whether directly or overtly. Instead, the DV experiences are made visible through many other indirect ways. The most common visible ways of DV and family violence symptoms being reported to GPs include minor injuries at different stages of healing, sleep issues, low self-esteem and other mental health problems [5,10,42].

Sleep difficulty is one of the most common problems among women who experience acts of violence [42,43,44]. However, this symptom is often associated with other women’s health issues, thus making it difficult to ascertain whether or not women are experiencing violence, assault or abuse. Many women who suffer from DV request prescriptions for sleep medication with synchronous symptoms of depression, anxiety and a desire or compulsion to self-harm [42]. It is challenging for GPs to initiate conversations about violence that women may face from their partner [42].

Mental health issues or psychological factors are key symptoms raised during GP visits by women who experience DV [34,42,45]. Most DV victims, whether they identify as such or not, attend their general practice regularly with comorbidities of mental and physical health issues [5,10,46]. Included studies reveal that female DV victims experience numerous mental health problems [3,23,34,45,47]. Generally, DV victims have very poor mental health and struggle to cope or function in everyday life [3,5,10]. Victims’ poor emotional well-being has a significant impact on their decision-making processes. For example, women visit GPs in a state of panic or anxiety, often having trouble communicating clearly at these times [34,45]. Women frequently want to seek professional support, yet they attempt to avoid doing so by convincing themselves that other people would perceive them as bad wives or partners [23,34]. Some women tend to think that they can manage DV situations by themselves; others think that the situations are temporary and will eventually resolve themselves, or that their abuser was going through a ‘bad phase’ or having a bad day [23,34]. Some victims “Dr shop” to avoid disclosing the real cause of their injuries and illnesses by seeing multiple GPs for a particular incident [22]. These mental factors often compound within the victims, thus preventing them from revealing their DV experiences.

Fear is a highly common characteristics among patients who visit GPs and other health services as the result of DV [5,10,47]. It has long been established that fear is a key barrier for women communicating abuse to primary healthcare providers [39]. Many women are unwilling to disclose what has happened, and most victims attempt to minimise the harmful incident [39]. Fears identified include fear of consequences from their partner, fear of more violence, fear of losing their partner and fear that they will not be believed [10,47].

Fear is a common psychological factor that patients experiencing DV exhibit, and while some of the causes of fear have been noted, an additional fear pertains to financial dependency [10]. According to the literature, victims’ financial situation is a crucial deciding factor in their willingness or confidence to disclose abuse [23,39,45]. Women who are financially dependent on their partners are afraid that they will be unable to survive without a source of income. Many abusers will work to ensure financial dependency as part of their abuse, coercion and control strategies. The abusers may do this directly by not allowing their partner to work, damaging their chances of working or forbidding contraception so that unplanned pregnancies make continued employment difficult [48,49,50]. Women’s income and motherhood status are also factors that prevented them from reporting the abuse to GPs or even leaving their partners [39,45].

### 3.3. Detection and Intervention in the Clinical Setting

The majority of female psychiatrists revealed that dealing with DV was not their responsibility or obligation [47]. DV is an issue that community health workers should handle rather than primary healthcare professionals or psychiatrists [47]. Male psychiatrists indicated that psychiatrists did not assist in identifying DV victims, but the appointment of a specific staff member would [47]. In addition, male psychiatrists reported that listening to and treating and dealing with female DV victims was a difficult and uncomfortable job because they felt guilty about the situations of their female patients [7,47].

GP centres, in theory, are intended to provide a safe and confidential way to disclose violence and abuse incidents [51]. These settings have unique characteristics for early abuse identification and are equipped in many ways to prevent DV through effective interventions and referral mechanisms [40]. Patient awareness of their GP’s availability, their trust in the healthcare practitioners and the potential feelings of comfortableness are the advantages of these settings as areas with great potential for effective DV intervention [40,51]. Evidence shows that a patient’s trust in GPs and GP centres is higher than in other types of primary health service providers. Patients also intended to use GP services more regularly than other types of health and social services, making them potent contact points for initiating DV conversations, such as what DV is and how to get help to escape abuse [10]. For these reasons, these clinical settings have been recognised as potentially efficacious settings for DV screening and identifying interventions [34] Many health professionals and health organisations recommend screening programs as an early-stage intervention method for readdressing and stopping DV and family violence [5,10].

The WEAVE randomised control trial (RCT) was one of the first studies to evaluate a DV screening-related program among women, with implications and suggested potential improvements for GP-based interventions [10,34,38]. The study helped to identify several ways of screening implementation and aiding effective intervention [34,38]. In addition, the MOVE study was the first RCT to determine the effectiveness of identifying intimate partner violence in a community-based nursing setting [32,52]. The MOVE was an intervention with a resource guide about intimate partner violence [32]. This study can be considered an effective step because it provided health practitioners in the clinical setting with relevant resources. According to the final MOVE intervention, the final results had no impact on regular reporting of DV cases or screening in referrals [32]. On the one hand, findings showed the same participants were involved in the intervention as a negative impact and noted a significant increase only in safety planning as a positive impact [32]. However, the study shed new light on self-completion checklists, which were effective in the clinical setting and contributed to a slight difference in establishing pathways to discuss DV experiences [32]. Overall, nursing-based models have proven to be effective in primary healthcare settings. However, the interventions or screening programs are required to be consistent with a victim’s safety planning, rather than simply asking direct questions to detect DV or family violence [32]. Safety of the victims who disclose abuse remains paramount during any screening or intervention activity, regardless of its point of administration or delivery [32].

Primary health professionals utilise numerous screening tools. The most popular screening tools are Hurt, Insult, Threaten and Scream [53]. Generally, this involves the screener asking the primary health service user questions during a screening process [34,54,55]. The screener has the opportunity to identify DV victims if they reveal their real condition, but most of the time, the victims do not do so [54,56]. In addition to the basic screening tools, brief health screening items, written or electronic identification methods, and in-person meetings have been reviewed and recognised as effective tools for reaching out to DV victims [36,52]. Risk assessment is another way of identifying family violence. It is mandatory in most primary health settings to implement a screening process before conducting a risk assessment [55]. During the risk assessment process, practitioners have the opportunity to ask more detailed questions [10]. Routine screening is another common strategy used in the primary healthcare setting [32,36]. Routine screening includes regular physical examination check-ups for skin conditions, sexually transmitted diseases and the eyes, as well as blood pressure levels [57]. Another approach that has shown some success in assisting women suffering from abuse is the ‘case finding’ or inquiry approach [32]. The case-finding approach can be applied in any DV situation, but healthcare workers should have relevant training to handle cases [32]. Social work professionals are more likely to use the case-finding approach, and in this scenario, public health professionals must work together with them. This method can map out victims’ personal experience in analysing DV situations [58,59,60].

Unfortunately, the reality at the pragmatic level differs from the theory [54,56,61]. Various complications have been found in screening programs, though screening is considered as a recognised way of identifying and preventing individuals from becoming victims or perpetrators. Moreover, screening for complex social phenomena in GP centres demonstrates a very low or limited data yield overall [32,36,52].

The screening process has several issues that needed to be rectified by the responsible authorities. Common claims include not interviewing in a private setting or space, having too many staff members involved in the screening process, the screener not being the same gender or race as the victims, the presence of the victim’s partner and age gaps between the victim and screener [54,56,61]. However, there is currently insufficient data or evidence to draw decisive conclusions about the effectiveness and potential for screening DV within GP practices and clinics [54,56,61]. The quality and outcome of DV screening programs and intervention processes depend on the timing and nature of the delivery of the questions by the healthcare provider to the patient [52].

Research has highlighted the complications and barriers to successful DV intervention and screening by GPs [5,34,36]. Firstly, the research acknowledges how profound the breakthrough can be for the patients and women who were disclosing their experiences for the first time. Due to the various reasons and fears that prevent women from revealing their living conditions, a GP’s chances of detection remain low overall. Establishing the necessary trust to reveal such experiences was profound and difficult for any health service provider to achieve [34,36]. Secondly, to be effective and safe, GP-based interventions in primary care settings should consider the different types and severity of abuse faced by women [10]. A common or universal general intervention is not feasible for the whole target population who have experienced DV. Nuanced responses and referrals are required to make discerning insights about the specific type of treatment and support the best matches for the experiences of each unique woman. Thirdly, there are still concerns that GP-based screenings and individual case data collection efforts do not always provide a complete and accurate account of the specific characteristics of the type and severity of harm [10]. One of the most frequently used data collection methods, self-reporting, has been discovered to have an inherent bias [5,10]. Response bias is a general complication within this type of data collection method [5,62]. Addressing all the characteristics of this highly diverse and vulnerable target population through a GP centre or individual clinic visits alone is a daunting and complex goal to achieve [34]. More research is needed on screening tools and strategies for the timing and nature of their delivery and administration if GPs are to achieve greater success in their efforts to assist victims and survivors to escape and fully recover from DV [38].

Finally, screening as an intervention tool for identifying DV remains questionable. It has several biases when used in the primary healthcare setting. It is therefore worthwhile to consider what is needed to generate more effective responses to DV in the primary healthcare setting.

### 3.4. Recommendations for More Effective Responses to DV in Primary Healthcare Settings

The literature widely acknowledges that improvements in the primary healthcare setting are much needed if they are to be better and more trusted places for victims of DV and other domestic abuse to seek assistance [37,40]. Beyond the internal reviews, evaluations of the screening tools and an increased capacity for GPs to be able to respond to patients suffering from DV are needed. DV experts and other community health service providers have weighed in to provide insights into how primary healthcare providers can better respond to this highly sensitive, diverse and complex social phenomenon.

When considering the macro level of the healthcare setting, one meaningful suggestion is that feminist-driven approaches need to be implemented in a primary healthcare setting to tackle gender imbalances in the clinical health context [63]. Literature suggests DV is a highly cultural and gendered issue that can be seen in many social structures [64]. This significant debate concerning power imbalances also exists in the primary healthcare setting and is rarely questioned by the responsible parties sitting upstream [65]. Gender inequality is considered as one of the key indicators in the primary healthcare setting that prevents effective decision-making for female DV victims [66]. Moreover, male dominance in the health sector is more likely to provide women with equal opportunities rather than equal rights, which can significantly impact victims or patients when they reveal their DV experiences [66]. However, male dominance and their hyper-masculine behaviour towards female victims compels victims to be male perpetrators’ perpetual bait [64,66,67]. These changes should occur at the ecological level, and they must be addressed for the overall well-being of women.

Female patients who visit GPs with DV comorbidities have several concerns at the micro level. One concern is the GP’s ‘communication style’. DV victims have revealed communication as a common barrier preventing them from disclosing their DV experiences [5,34,51]. Australian studies have revealed that most victims would like to see some improvement in their GP’s current communication style, which they claim is not conducive to feelings of trust and equality, inhibiting them from sharing their intimate life details [34,40]. Evidence demonstrates that mutually supportive communication supports victims to increase their self-confidence to discuss the topic with their GP [34,40]. This is a common desire among patients who use mental health services [47]. Many women who seek mental healthcare support report that they require their GPs to take a similar approach in terms of communication sensitivity in these spheres if they were to open up and share their stories [47]. Victims want to feel safe, which can only be achieved if the GP’s communication style leads them to trust that this healthcare professional will not perceive them as being guilty for creating a situation that harmed their physical and mental health [68]. Primary healthcare providers require greater DV training and sensitive doctor–patient communication for these women to feel confident that the primary healthcare providers are competent in assisting them in their respective abusive situations [40].

Despite the reported competency gaps, the majority of healthcare professionals, including psychologists, psychiatrists and GPs, recognise DV as a serious health problem with huge social and economic costs to the country [7]. Proper training in sensitively screening victims will support healthcare professionals to identify DV victims [7]. However, this intention to improve skills and training in this area has not yet translated into a reduction in the skill gap of DV-based competence in primary healthcare professionals. Upskilling health practitioners should be considered as a given [7]. Nurses have reported feeling that they are not sufficiently aware of how DV works in terms of coercion and control, nor the inequities and power imbalances that drive and sustain it [69]. Insufficient skills and training to identify the signs of DV among healthcare professionals is reportedly common and covers the areas of communication skills, practical knowledge in DV, self-confidence, theoretical knowledge, skills to use relevant educational materials, proper knowledge of referral services, training in preparedness to face victims, skill development, identifying victims’ behavioural patterns and accurate screening skills [7,34,41,69]. There is no current evidence demonstrating that sufficient training or resources are available for health staff to increase the skills and knowledge they need to gain the self-confidence and nuanced skills to identify DV safely in clinical settings [7,47,69,70,71].

Self-efficacy, self-confidence and self-esteem are reportedly key characteristics needed in primary healthcare professionals to work more effectively with DV victims and survivors [71]. Studies reveal that their perceived lack of self-efficacy (e.g., confident in being able to support victims and perpetrators in future nursing practices) is a main barrier preventing them from reaching out to potential sufferers and engaging in conversations with their patients about domestic abuse [71]. Low self-esteem in relation to these skills reportedly generates confusion and consequently unsuccessful assessments of their patients and low-quality reporting of cases [71]. Findings from the Australian context confirms that healthcare professionals are not confident in DV screening, identifying victims or referring victims to relevant support [7,69]. GPs’ low confidence rates in their ability to properly and effectively assist their patients with DV combined with patient fear and low trust in GPs as people with whom they are likely to share their experiences, invariably results in faulty reports or incomplete assessments and low satisfaction for both GPs and patients [47]. For example, “*People (staff) are hesitant because they do not feel confident, they do not feel it is their job; they think that somebody else is better equipped to do it*” (P12, male, psychiatrist) [47]. The most common answers from nurses and midwives are the lack of privacy, knowledge, education and relevant resources [69]. Due to a lack of preparedness, nurses feel bad dealing with DV victims [71].

According to health professionals, they face numerous barriers when dealing with DV victims. Insufficient family violence patient resources, not having enough education resources, victims’ uncertainty about their situation, lack of education and skill-based knowledge to deal with DV victims and not having specific training based on DV or family violence are most common critical issues [7].

Experts and scholars say that time is a crucial factor within the general practices. The duration of a GP consultation session is a decisive determinant in screening for family violence [7,68]. Studies reveals that 15 min of GP appointments are not sufficient to discuss DV experiences [7,22]. They suggest this issue is a sensitive concern [7]. During a general consultation is not the right time to discuss those experiences due to time barriers and heavy GP workloads [7]. The fact that GPs are unable to use this time to discuss DV experiences of their patients has been a significant issue for a long time [22,52]. There is considerable discussion on healthcare professionals’ attitude, workloads, lack of training, inadequate consultation time, insufficient resource support and victims who present to the clinical health practices with their partners [52]. There is also an issue of health professionals’ understanding their role: “Though I wanted to help victims, that is not my job” as one health professional described it [68]. These characteristics of general practices exist as barriers to identifying the signs of DV within the general practice setting.

Interventions and screening programs present as another area for improvement. Professionals have identified several improvements for implementing effective interventions in the primary healthcare setting [34,71]. For instance, DV interventions should address the victim’s emotional needs [71]. Skill development should be compulsory to help practitioners identify the early symptoms DV within the primary healthcare setting [69]. Scholars present that most of the DV interventions are ineffective and do not provide the supporting environment to allow victims reveal personal experiences [68]. Almost all the nursing interventions concentrate on screening programs [68]. The healthcare system should find a more responsive service rather than screening [68]. Another issue that remains to be solved is the relationship between healthcare professional and the victim [68]. The tension between them leads healthcare workers to judge victims as abnormal and unacceptable [68]. For example, “*You, you talk to the patient, and you know, you get their story, “Oh, OK, yeah, you know that’s terrible”. Then, you talk to the psych services who know this patient very well and they give you the real story and it is completely different. You have been thrown off track by this patient*” (Sam) [68]. This kind of tension in the healthcare field needs to be solved to address the issue of DV [68]. To provide an effective response in primary healthcare services, it is imperative that professionals understand women’s thinking and their experiences [68].

## 4. Discussion

This scoping review has located and discussed the most relevant articles on the reported barriers faced by Australian and New Zealand women experiencing DV in sharing their experiences with primary healthcare providers. Several journal articles, government organisations, non-government organisations and the Department of Health focus on the statistical data surroundings this serious public health concern [3,7,18,19,20,22,23,24,25,26,29,59]. The reason for this is that the incidence and prevalence of DV cases are gradually increasing—a fact that these responsible bodies are acutely aware of.

Within the primary healthcare settings and specifically in GP settings, it is a challenging task to identify DV victims unless they are willing to reveal their experiences of harassment, physical harm or sexual harm [10,54,56]. Victims are more likely to present with various other ill-health symptoms, such as sleep difficulties, mental health issues, injuries, fears or psychological factors that have been shown to be hidden and directly related to DV cases [5,10,34,39,42,43,44,45,48,49,50].

The review findings show that interventions implemented in the Australian primary healthcare and clinical settings to identify DV are not sufficient and are currently not operating in a way that achieves effective outcomes [5,32,34,38]. Additionally, DV screening programs are the most prominent intervention type within the Australian primary healthcare sector. Existing implementations are subject to several complications, including issues concerning self-completion surveys, self-reporting tools, selection bias in RCTs and not revealing the truth because of the fear of more intimate partner violence [5,10,32,34,36,38,56,59,60]. Despite the interventions, the majority of healthcare professionals are not aware of DV situations, victims, the signs or do not know how to react to the cases [10,34,47,51,68]. Healthcare professionals are in need of upskilling their knowledge, self-confidence, theoretical background, educational support and skill development regarding this social phenomenon.

Finally, gender imbalance and inequality between male and female health professionals within the primary healthcare setting appears to be a significant indicator of the quality of the health services provided within the primary healthcare settings and that offered by primary healthcare professionals [63,64,65,66,67]. Globally recognized strategies to reduce gender-based power differences at work, such as affirmative action, gender mainstreaming, gender equity training, and the encouraging of women into medicine degrees over nursing degrees is required to redress this imbalance in healthcare systems. This scoping review has identified that power imbalances exist not only in personal relationships between two human beings but also across medical relationships [66].

### Limitations

There were a few limitations to this scoping review. To examine the topic, a broad range of journals and databases were searched. It was not the aim nor the intention to undertake a systematic literature review, and as such, the documents we located as a result of the search terms and syntax we employed did not yield a complete set of all possible articles on this topic. Future systematic reviews could specifically include a focus on words such as ‘symptoms, comorbidities, detection, and interventions’, for example. Search strategies were developed that reflected the immediate aims and objectives of the research, and provide a snapshot of what research is available to address a specific set of questions. However, the articles located were indeed able to provide the findings we needed to provide answers commensurate with the aims of this review. Moreover, the scoping review was limited to articles in the English language.

## 5. Conclusions

This scoping review collated the current evidence available within the scope of our search methodology on the many reasons that DV victims are reluctant to openly discuss their DV experience at the primary healthcare level. According to the perspective of Public Health, primary healthcare professionals play a vital role in preventing and managing DV against women, however, this is currently undermined due to a range of barriers to communicating situations and symptoms to clinicians in private settings A core finding emerging from the review was that the current power imbalance between male and female staff across allied and clinical health sectors be remedied. This issue has become a staple problem in the social structures and health settings throughout the decades and is particularly sensitive in the realm of DV detection and interventions. Moreover, this power imbalance is considered as a general and normal occurrence within the Australian primary healthcare setting, which is highly problematic. It is of concern that this power imbalance seeps into any social structures, given that these women already face massive power imbalances in their day-to-day lives.

The review also concluded that while screening is the principal intervention tool used to identify DV victims within GP centres and other primary healthcare service providers, it is not always confidently applied by practitioners nor sought out by DV victims during visits. Innovative interventions are needed within these settings, such as effective and more nuanced, or sensitive DV screening tools, risk assessments and case study findings to generate ways in which a rapport between GP and patient can be generated and protected during screenings. Accurate, sensitive, and safe screening can support health providers to identify victims at the right time [12]. GPs also need to become far more educated regarding the clusters of comorbidities that typically accompany a DV victims health report. While the DV itself may not be communicated in clinical settings, all healthcare providers need to be educated on the ‘red flags’ such as sleep problems, anxiety, and substance use that often point to an underlying set of DV conditions. On the other hand, victims need to be made much more aware of benefits of screening programs and other DV prevention tools. Victims are often not aware of what support is available for them and primary healthcare providers often fail to refer victims to such support.

Further research is needed to collect more accurate and reliable data regarding disclosure in healthcare settings. Specifically, there is a concerning deficiency in population-based studies and research, which could be the most effective for researchers, scholars, public health practitioners, policy advocates and primary healthcare service providers. Health policymakers must be aware of equal rights with equal opportunities for female workers in the primary healthcare setting. Policymakers must also pay attention to public health norms, due to the importance of women’s overall health consequently reflecting the health of the country’s future generations. Advocating for changing the social structure is of the utmost importance to ensure both male and female professionals are present at the first layer of Australian healthcare. This should be considered as a mandatory requirement to empower women.

## Figures and Tables

**Figure 1 healthcare-11-02486-f001:**
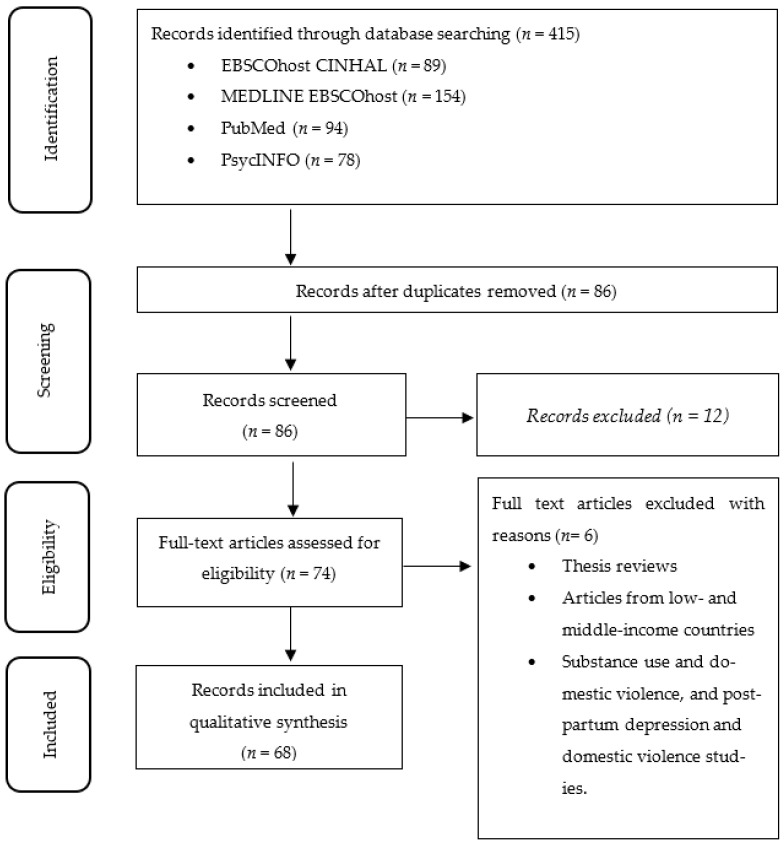
The flow diagram for the selection process and reasons for exclusion of studies.

## Data Availability

Not applicable.

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
