# Peer review of "Barriers Faced by Australian and New Zealand Women When Sharing Experiences of Family Violence with Primary Healthcare Providers: A Scoping Review"

_healthcare, 2023, doi:10.3390/healthcare11182486_

Round 1

Reviewer 1 Report

Please see my comments attached

Author Response

Thank you so much for this feedback. We genuinely feel that your suggestions to better align the paper, and to more tightly generate its narrative or thesis has resulted in significant edits and revisions that have improved our paper. 

I will attempt to supply two versions of the paper with your revisions included; one with the track changes so that you can see where all the examples you requested have been provided, and more importantly, the big changes to the introduction, research aims/questions, and discussion and conclusions have been made, and the other with all track changes accepted to give you a sense of the paper in its new form.

We worked hard to be very clear about the focus on the three core aims of the scoping review:

(i) What are the reasons DV victims do not disclose to GPs and primary healthcare professionals? (ii) What are the comorbidities that DV patients present with? (iii) what are the current methods of detection and interventions in clinical settings. The objective was to combine the findings to provide recommendations for both researchers and clinicians regarding more effective responses to DV.

We have been iterative in our approach to editing the paper to reflect these aims and the findings from the introduction to the concluding comments. 

We hope that you approve of our revisions and changes in accordance with your feedback

Reviewer 2 Report

11.       On page 1, line#39, “USDOH” should be replaced with “USDHHS” or “USDHS”.

2.       On page 9, line#379, authors will clarify healthcare professionals’ “lack of self-efficacy” with examples. Then, line#387, authors will also clarify healthcare professionals’ “low self-esteem” with examples.

3.       On page 9, line#420 , authors will provide examples for “tension between them leads healthcare workers to judge victims as abnormal and unacceptable”.

4.       On page 10, line#469 and #470, authors will give examples how to address “power imbalance between males and females in the healthcare sector”.

Author Response

  1. On page 1, line#39, “USDOH” should be replaced with “USDHHS” or “USDHS”

Response(s):  changed to USDHHS in document – line 39 and 41.

  1. On page 9, line#379, authors will clarify healthcare professionals’ “lack of self-efficacy” with examples. Then, line#387, authors will also clarify healthcare professionals’ “low self-esteem” with examples.

Response(s): Examples were provided in the document.

For line #379 (now line#382), changed was made in bold sentence: “Studies reveal that their perceived lack of self-efficacy (e.g., confident in being able to support victims and perpetrators in future nursing practices) is a main barrier preventing them from reaching out to potential sufferers and engaging in conversations with their patients about domestic abuse”.

For line #387 (now line#393), example was included: For example, “People (staff) are hesitant because they do not feel confident, they do not feel it is their job; they think that somebody else is better equipped to do it” (P12, male, psychiatrist) [45] 

  1. On page 9, line#420, authors will provide examples for “tension between them leads healthcare workers to judge victims as abnormal and unacceptable”.

Response(s): Example was provided in document.

For example, “You, you talk to the patient, and you know, you get their story, “Oh, OK, yeah, you know that’s terrible”. Then you talk to the psych services who know this patient very well and they give you the real story and it is completely different. You have been thrown off track by this patient” (Sam) [72]. – line #427

  1. On page 10, line#469 and #470, authors will give examples how to address “power imbalance between males and females in the healthcare sector”.

Response(s): The following added from line#463 to #467: ]. Globally recognized strategies to reduce gender-based power differences at work, such as affirmative action, gender mainstreaming, gender equity training, and the encouraging of women into medicine degrees over nursing degrees is required to redress this imbalance in healthcare systems. 

Reviewer 3 Report

Thank you for the opportunity to review this scoping review on domestic violence (DV) barriers and experiences within the primary healthcare settings and to primary healthcare professionals by Australian women.

Suggestions to improve the manuscript:

Title: as only Australian women is mentioned, then data from New Zealand women could not be included, so align title with search strategy

Abstract: page 1, line 20, are the recommendations 'further research and interventions needed' as this is not so helpful to readers; make more specific, e.g. referral processes to whom? this information does not match conclusion of manuscript so align

Introduction:

- page 1, lines 30 and 31, reference 3 is too old, update with for example, WHO Violence Against Women Prevalence Estimates, 2018 released 9 March 2021, https://www.who.int/publications/i/item/9789240022256

- page 1, line 32-33, move to the end of the 2nd paragraph to first state the consequences to victims before stating that also children and pets may be involved

- page 1, add to line 36 the targets for the new SDGs, to show still relevant in international policy and increased upon in visibility

- line 87 to 88 is repeated in section 2.1 so keep in only one place

- line 91, typo in comorbidities, run spellcheck prior to submission

Methods

- 2.1 has to include a research question for each of the aims (i-iii) stated in paragraph above as otherwise not aligning

- 2.2 again here aim is repeated, remove

- 2.2 add date of search, for example studies were included starting what year till what endpoint / provide appendix file with exact search strategy in compliance with PRISMA ScR best practice so can be repeated/checked

- add PRISMA ScR checklist as appendix/suppl material and mention in methods text that this was followed (http://www.prisma-statement.org/Extensions/ScopingReviews?AspxAutoDetectCookieSupport=1 )

Results

- this section can only include references that met criteria for the scoping review, all other information pertaining to any other references for comparison should be moved to the discussion section. For example reference 67 is Australian so is it part of the scoping review? while ref 68 belongs to discussion as not part of the review, correct?

- Add a Table 1. Characteristics of included studies, with the information on all of the 68 included references like other scoping reviews typically due, so that readers can see this at a glance

Discussion

- page 10, line 427, if New Zealand was included, must be mentioned here

- in this section it is the authors' duty to compare and contrast their findings to other research, so please shift any information, references from results that were not from your scoping review to this section to compare with results from other countries--how similar and how different, or compare to low income countries--how similar and how different, etc. This is necessary for all three parts of the results.

4.1 Limitations

- add review was not systematic, thus no quality assessment made of the journal articles regarding bias, etc.

- line 460-461, restrictions to balance the literature, please explain in more detail as this is not clear to the reader

References:

- place an asterisk in front of each reference that was included in the scoping review, so easy to identify

Minor edits only.

Author Response

Title: as only Australian women is mentioned, then data from New Zealand women could not be included, so align title with search strategy

Response(s): Title changed to reflect search strategy.

Abstract: page 1, line 20, are the recommendations 'further research and interventions needed' as this is not so helpful to readers; make more specific, e.g. referral processes to whom? this information does not match conclusion of manuscript so align.

Response(s): Amended: This scoping review provides formative evidence that more accurate and reliable data regarding disclosure in healthcare settings be collected, that gender power imbalances in the health workforce be redressed, and that advocacy of gender equality and the change of social structures in both Australia and New Zealand remain the focus of reducing DV in these countries. 

Introduction:

- page 1, lines 30 and 31, reference 3 is too old, update with for example, WHO Violence Against Women Prevalence Estimates, 2018 released 9 March 2021, https://www.who.int/publications/i/item/9789240022256.

Response(s): I am not sure whether by updating the reviewer asked us to remove the reference 3. Reference 3 is included in the result section and I checked the paper was conducted in/or using Australian data.

- page 1, line 32-33, move to the end of the 2nd paragraph to first state the consequences to victims before stating that also children and pets may be involved

Response(s): Line 32-33 were moved to the end of the 2nd paragraph. Currently, the new line number are 46-47. (J-A, I need your help whether to make the first paragraph balance (or combine) with second paragraph in terms of number of lines since the first paragraph is only 4 sentences).

- page 1, add to line 36 the targets for the new SDGs, to show still relevant in international policy and increased upon in visibility.

Response(s): Changed was made in document on line 39.

Eradicating violence against women was included in the United Nations’ Millennium Development Goals (in 2000) as well as in the Sustainable Development Goal 5 (Gender Equality) (in 2015) [8,11].

- line 87 to 88 is repeated in section 2.1 so keep in only one place

Response(s): I deleted line 87-88. However, I am not sure whether we should remove these lines since section 2.1 is our research question

- line 91, typo in comorbidities, run spellcheck prior to submission

Response(s): Changed in the document as suggested. 

Methods

  • 2.1 has to include a research question for each of the aims (i-iii) stated in paragraph above as otherwise not aligning

Response(s): Research questions included in 2.1 as recommended.

- 2.2 again here aim is repeated, remove

Response(s): Changed in the document as suggested.

- 2.2 add date of search, for example studies were included starting what year till what endpoint / provide appendix file with exact search strategy in compliance with PRISMA ScR best practice so can be repeated/checked.

- add PRISMA ScR checklist as appendix/suppl material and mention in methods text that this was followed (http://www.prisma-

Response(s): This was not a Systematic Literature Review, and as such we did not follow PRISMA ScR. This is clearly outlined in our introduction and methodology. 

Results

- this section can only include references that met criteria for the scoping review, all other information pertaining to any other references for comparison should be moved to the discussion section.

Response(s): Study 68 removed from findings section and into discussion section as recommended.

- Add a Table 1. Characteristics of included studies, with the information on all of the 68 included references like other scoping reviews typically due, so that readers can see this at a glance

Response(s) - This is fitting with a more traditional Systematic Literature Review. This was a scoping review written up as a comprehensive discussion of the current available research pertaining to our specific research questions

Discussion.

- page 10, line 427, if New Zealand was included, must be mentioned here

Response(s) - edited to include NZ as recommended.

4.1 Limitations

- add review was not systematic, thus no quality assessment made of the journal articles regarding bias, etc. 

  • line 460-461, restrictions to balance the literature, please explain in more detail as this is not clear to the reader.

Response(s): All of 4.1 revised for clarity and specificity as recommended. 

References:

- place an asterisk in front of each reference that was included in the scoping review, so easy to identify

Response(s) - we will need to contact our first author who is currently overseas to complete this. We are unsure that this is necessary given the scope and clearly outlined limitations of this review. 

Round 2

Reviewer 3 Report

Thank you for improving the manuscript for publication. 

Minor edits to make:

- abstract: to be consistent with the manuscript, replace 'reveal' with disclose and use only this word and not report (replace in 3.1, line 384) as this is a different terminology, namely to report to police or to an authority; line 22, remove and and start sentence with 'These..'

- 2.1. This heading should read questions, plural since more than one and either add Objective to heading or remove the objective at end of paragraph (preferable) and make the recommendations into a research question. Also ii should include 'symptoms and comorbidities' as your results include both and be clear on their definition, although it is not clear in your data search strategy that you would have found all of them since not a word you searched for. The same critique holds for the search for detection and interventions. Please comment on this in the limitations section.

-3.4 replace 'What is needed' with Recommendations to align with 2.1 / the text is hard to follow so suggest adding subheadings to help the reader follow, e.g. authors start out with macrolevel, then if used, next section should refer to micro and package text accordingly

- Further research should be moved to discussion, 4.2

- 4. add references to justify lines 488 to 491 that these are evidence based

Fine, only minor review needed.

Author Response

  • abstract: to be consistent with the manuscript, replace 'reveal' with disclose and use only this word and not report (replace in 3.1, line 384) as this is a different terminology, namely to report to police or to an authority; line 22, remove and and start sentence with 'These..'
  • Response: Great advice re word choice. All replaced as advised. And the sentence now starts with 'These'.
  • 2.1. This heading should read questions, plural since more than one and either add Objective to heading or remove the objective at end of paragraph (preferable) and make the recommendations into a research question. Also ii should include 'symptoms and comorbidities' as your results include both and be clear on their definition, although it is not clear in your data search strategy that you would have found all of them since not a word you searched for. The same critique holds for the search for detection and interventions. Please comment on this in the limitations section.
  • Response: Done.

-3.4 replace 'What is needed' with Recommendations to align with 2.1 / the text is hard to follow so suggest adding subheadings to help the reader follow, e.g. authors start out with macrolevel, then if used, next section should refer to micro and package text accordingly

Response: Done. 3.4 question mark removed to be clear that it is addressing the recommendations as outlined in 2.1

- Further research should be moved to discussion, 4.2

  • 4. add references to justify lines 488 to 491 that these are evidence based
  • Response: Reworded to note that this was not a comprehensive systematic literature review.